# Making MALDI-TOF MS for entomological parameters accessible: A practical guide for in-house library creation

Jonathan Karisa[1]*, Mercy Tuwei[1], Kelly Ominde[1], Brian Bartilol[1], Zedekiah Ondieki[1], Haron Musani[1], Martin Rono[1], Charles Mbogo[1], Philip Bejon[1,2], Joseph Mwangangi[1], Caroline Wanjiku[1], Marta Maia[1,2,3]*

1 Kenya Medical Research Institute, Wellcome Trust Research Program, Kilifi, Kenya, 2 University of Oxford, Centre for Global Health and Tropical Medicine, Oxford, United Kingdom, 3 Pwani University, Kilifi, Kenya

* karisajonna@gmail.com

## Abstract

Matrix-assisted laser desorption-ionisation time of flight mass spectrometry (MALDI-TOF MS) is a powerful analytical method that has been used extensively to identify sample ions of complex mixtures, and biological samples such as proteins, tissues and microorganisms. MALDI-TOF MS has revolutionised clinical microbiology with accurate, rapid, and inexpensive species-level identification of microbes. MALDI-TOF MS technology generates spectral signatures and matches them to a library of similar organisms using bioinformatics pattern matching. The use of MALDI-TOF MS for entomological samples has been explored by multiple groups with proven efficacy at differentiating between closely related species, as well as detecting pathogens in different vectors. The low cost per sample processing, rapid turnaround and robustness are attractive for surveillance of vector control programs. Libraries are built in-house for institutional usage, although a multi-user platform with sharing of spectra and data would be attractive. Only a few studies have strived to make their libraries publicly available. Here, we outline a stepwise approach for creating an in-house MALDI-TOF MS library and subsequent query, using malaria vector species identification as a case study for entomological samples. A protocol and video of the methodology are also shared. Moreover, the libraries related to this publication have been deposited in public repository (https://doi.org/10.7910/DVN/VYQFNO37) for anyone with MALDI-TOF MS equipment to adapt.

## Introduction

Vector control remains a cornerstone of malaria prevention, with vector surveillance playing a critical role in monitoring and evaluating control and research programs. However, for surveillance to be truly informative, high-quality data must be collected

**Data availability statement:** Sample data are fully available without restriction on Harvard Dataverse - https://doi.org/10.7910/DVN/VYQFNO

**Funding:** MM - Investment 057212 - Bill and Melinda Gates Foundation - USA MR - FLR_R1_190497 - The Royal Society FLAIR fellowship grant- UK The funders had no role in study design, data collection and analysis, decision to publish, or preparation of the manuscript.

**Competing interests:** No potential conflict of interest was reported by all the authors.

consistently across space and time, extending beyond simple vector densities and morphological identification. A major challenge is that molecular assays (essential for understanding vector species composition) are prohibitively expensive when applied to thousands of mosquitoes annually. As countries progress toward malaria elimination, gaining deeper insights into vectorial systems and transmission patterns becomes essential for designing end-game strategies. This underscores the urgent need for cost-effective tools that enable large-scale sample processing, ensuring robust data collection to drive informed decision-making in malaria control and elimination efforts.

MALDI-TOF MS represents one such tool. The technology has revolutionised diagnostics in the field of clinical microbiology allowing for faster turn-around time from sample preparation to obtaining precise and accurate results at a lower cost per sample [1–4]. Recently, MALDI-TOF has found valuable application in entomological surveillance. Numerous studies have assessed its performance, accuracy, and reliability for mosquito species and sub-species identification [5–8]. MALDI-TOF MS has also been evaluated for the identification of other arthropods, including Drosophila, [9,10], fleas [11], lice [12], ticks [13–16], Ceratopogonides (biting midges), and sand flies [17–19]. It has also has also been used to differentiate between infected and non-infected ticks [15,16,20]; *Plasmodium*-infected and non-infected *An. stephensi* [21] and Wolbachia-infected and non-infected *Aedes aegypti* [22]. MALDI-TOF MS has also been shown to distinguish between/among blood meals from different hosts [5,23–25], age grading of malaria vectors [26] and insecticide resistance profiling [27].

Creating in house libraries can be challenging and often the number of institutions that have access to the instrument but that use it beyond microbiology is limited. We aim to demystify the process and share a detailed workflow for creating a MALDI-TOF MS library using malaria vector species identification as a case study. When well optimised, the spectra obtained from a single run can be used to develop other libraries and query other parameters including, for example, infectious status [21], age/parity [26], presence endosymbionts such as *Microsporidia MB* or *Wolbachia* [22], and potentially insecticide resistance [27]. Blood-engorged mosquito abdomens can be used for differentiation of blood meal sources [5].

The workflow involves a series of steps that begins with dissection of mosquitoes into three anatomical compartments: legs and wings, cephalothorax, and abdomen. Dissection is followed by protein extraction and acquisition of spectral profiles. Guided by results from gold standard tests (ELISA, PCR, taxonomy), a sub-set of high-quality spectra is then selected and used for training the machine algorithm and create a reference database (spectral library) (training dataset).

A high-quality spectrum should have a high number of well-defined, intense peaks with good signal-to-noise ratio, consistent peak positions across replicates, minimal background noise, and a set of prominent "marker peaks" specific to the analysed molecule, allowing for reliable identification and comparison against a reference database. The remaining spectra are used to determine the accuracy of the reference database in identifying unknown samples (validation dataset). The match level

between the unknown sample and the reference database is estimated using the log score value (≥1.8 is considered a reliable identification). Once libraries are created these can be continuously improved by inclusion of additional reference spectra capturing potential variation between strains due to geographical difference brought about by genetic variations limited arising from limited genetic exchange/gene flow.

Here we share the methodological approach to creating MALDI TOF MS libraries for entomological samples using malaria vector species identification as an exemplary case study. Moreover, the libraries related to this publication have been deposited in public repository (https://doi.org/10.7910/DVN/VYQFNO37) for anyone with MALDI-TOF MS equipment to adapt.

## Materials and methods

### Mosquito sampling

Mosquito samples were collected from 2019–2022 in three distinct ecological zones with different species composition viz., Kilifi, Taita-Taveta and Kwale Counties of Kenya. A description of mosquito sampling and processing is detailed in Karisa et al [5].

### Ethical consideration and approvals

This study did not involve human samples; only mosquitoes were collected and analyzed. However, since collections were conducted at the household level, ethical approval was obtained from the Kenya Medical Research Institute Scientific and Ethics Review Unit (KEMRI SERU 4945) before commencing the study. Several protocols were used during the mosquito collection exercise. Informed consent was sought from all participating households. In some cases, verbal consent was obtained, while in others, written consent or permission was provided in the presence of local administration before setting up traps for mosquito collection.

### Sample preparation and gold standards

The step-by-step protocol of the process here briefly described, is also available on protocols.io (dx.doi.org/10.17504/protocols.io.n2bvjdy3xvk5/v1) and is included as a supplementary file with this article (S1 File). Briefly, Anopheles mosquitoes were individually dissected into different body parts: legs, wings, cephalothorax and abdomen. The legs and wings or one half of the cephalothorax were used for species identification by PCR as the gold standard reference [28–30] for the classification of MALDI-TOF MS spectra. The remaining half of the cephalothorax was homogenized, proteins were extracted and loaded onto a steel MALDI-target plate (Bruker Daltonics) as described earlier [5]. The prepared plate was then introduced into the Microflex machine (Bruker Daltonics) for spectral acquisition.

### MALDI-TOF MS processing

The stepwise process of spectra acquisition, pre-processing, library creation and query (validation) has been previously described [5]. Furthermore, the step-by-step protocol of the process is also available on protocols.io (dx.doi.org/10.17504/protocols.io.n2bvjdy3xvk5/v1) and is included as a supplementary file with this article (S1 File).

## Results

### Mosquito molecular identification and spectra acquisition

A total of 2,332 mosquitoes, collected in Kenya [5], were morphologically identified as either *Anopheles gambiae* s.l or *Anopheles funestus* s.l and used for development of a MALDI-TOF MS species identification library. A total of 1,971 (85%) of the mosquitoes were accurately identified to species using molecular tools as quality control of reference library. The remaining 15% did not render sufficient DNA for PCR or were found not to be an anopheline by sequencing. Out of the

 

1,971 samples, various mosquito species were identified, including: *An. gambiae* s.s, *An. arabiensis, An. merus, An. quadriannulatus, An. funestus* s.s, *An. rivulorum*, and *An. leesoni,* making up the majority [5]. Of the 1,971 samples 5.2% produced poor-quality spectra and were therefore excluded, leaving a total of 1,782 for database creation and query (validation).

## Reference spectral library creation

Reference libraries were created with 87 well characterized sample spectra belonging to *An. gambiae* (*An. gambiae* s.s – 10, *An. arabiensis* – 26, *An. merus* – 19, *An. quadriannulatus* – 14) and *An. funestus* (*An. funestus* s.s – 6, *An. rivulorum* – 6 and *An. leesoni* – 6) complexes were used for database creation. The libraries related to this publication have been deposited in public repository (https://doi.org/10.7910/DVN/VYQFNO37) for anyone with MALDI-TOF MS equipment to adapt.

## Reference spectral library query

Out of all the mosquito analysed by MALDI-TOF MS, 1,712/1782 were correctly identified with an accuracy of 96.1% [5]. The database was particularly precise in distinguishing *An. gambiae* and *An. funestus* complex members which clustered into three distinct branches: i) *An. vaneedeni, An. funestus* and *An. parensis clustered, ii) An. gambiae* complex, and iii) *An. rivulorum* and *An. leesoni* (Fig 1).

However, a few samples had low LSV values, suspected to be due to either protein degradation, physiological status or residual blood meals in the head and thorax (Fig 2). The application of the weighted LSV approach on the *An. gambiae*

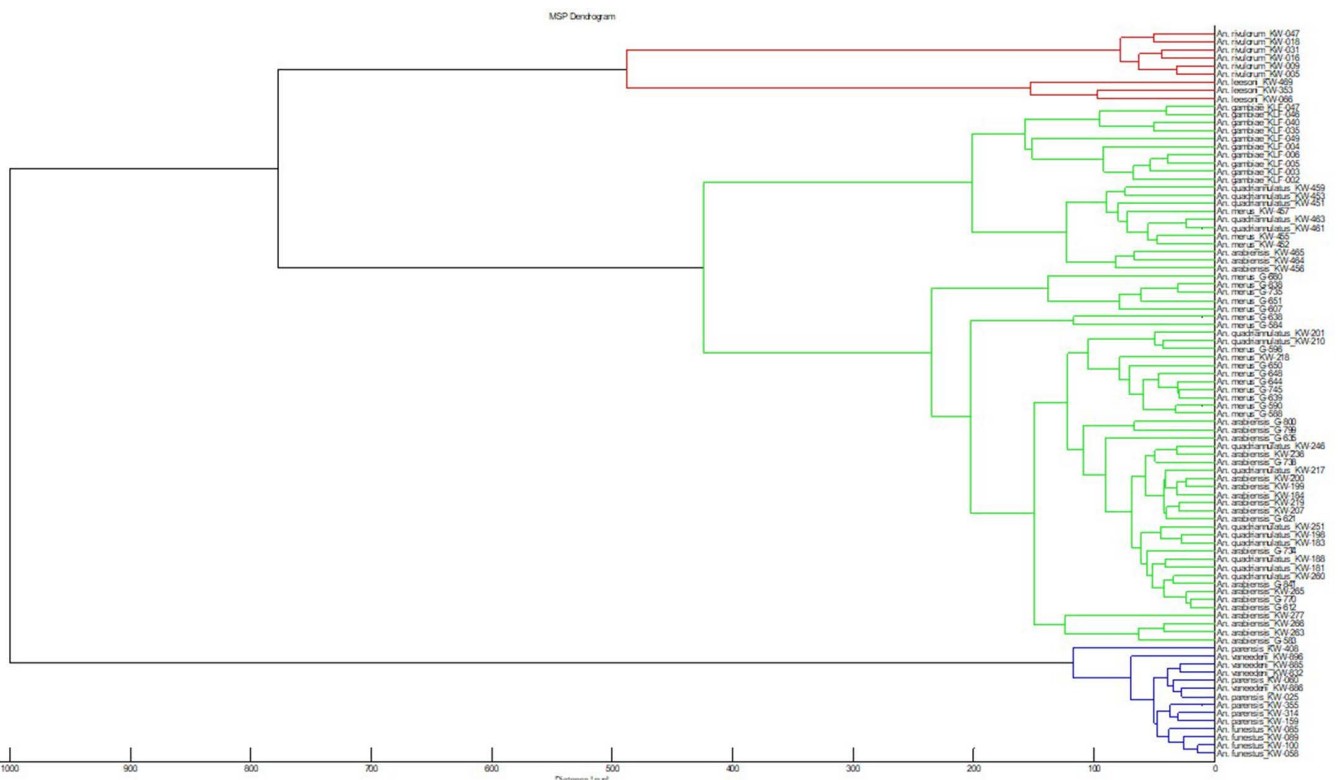

**Fig 1. MSP dendrogram of MALDI-TOF MS spectra used for database creation for species identification and *An. vaneedeni* and *An. parensis*** [5].

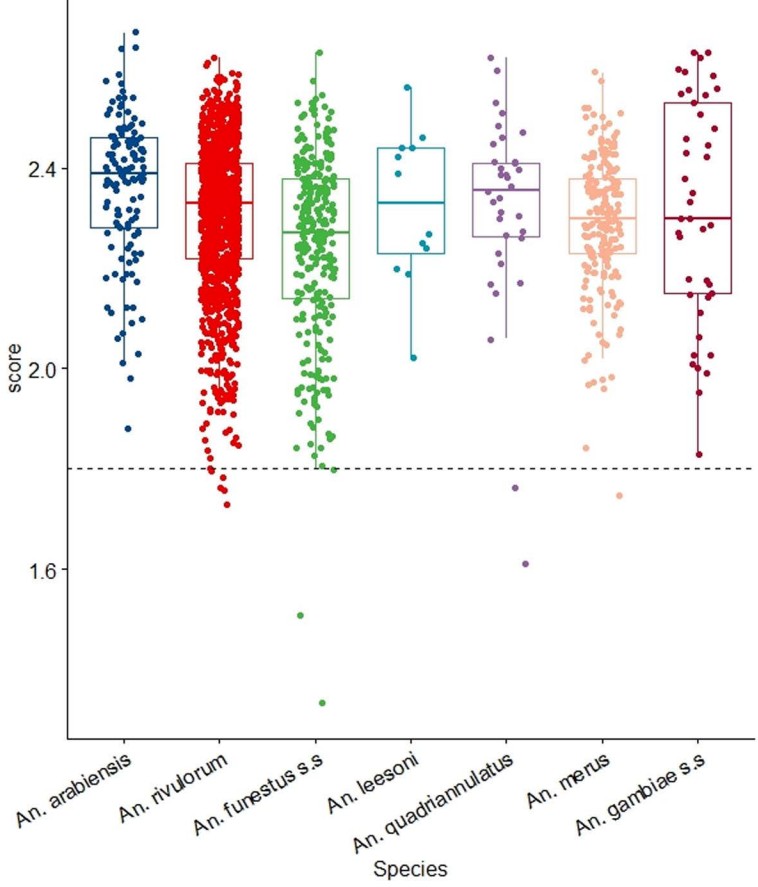

**Fig 2. LSVs obtained after a database query with MS spectra belonging to members of *An. gambiae* and *An. funestus* complexes.** Horizontal dashed lines represent the cut-off point for reliable identification (LSV > 1.8). *Abbreviations*: **A**.U., arbitrary units, *An*., *Anopheles*, s.s, sensu stricto, LSV., Log Score value [5].

s.l samples that had ambiguous identification during querying, resulted in correct species identification congruent with the molecular analysis results, further improving the accuracy of species identification to 97.5% [5].

### Impact of preservation method on mosquito identification and LSV distribution

With a well curated, preservation methods do not significantly impact on mosquito identification and LSV distribution as we noted that there was no significant difference (Kruskal-Wallis test, p = 0.19) in the median LSV among different methods of preservation (Fig 3).

### Discussion

This article outlines a comprehensive procedure for utilisations of MALDI-TOF MS for malaria vector species identification. The accuracy in distinguishing between sibling species of the two Anopheles' species complex (*An. funestus* s.l and *An. gambiae* s.l) demonstrates the discriminative power of MALDI-TOF MS. In the context of surveillance, the wide range of species representation should reflect as much as possible the diversity of mosquitoes in each study area. Different mosquito species play varied roles in disease transmission and have different feeding and resting proclivities. The composition of vectors paints a picture of

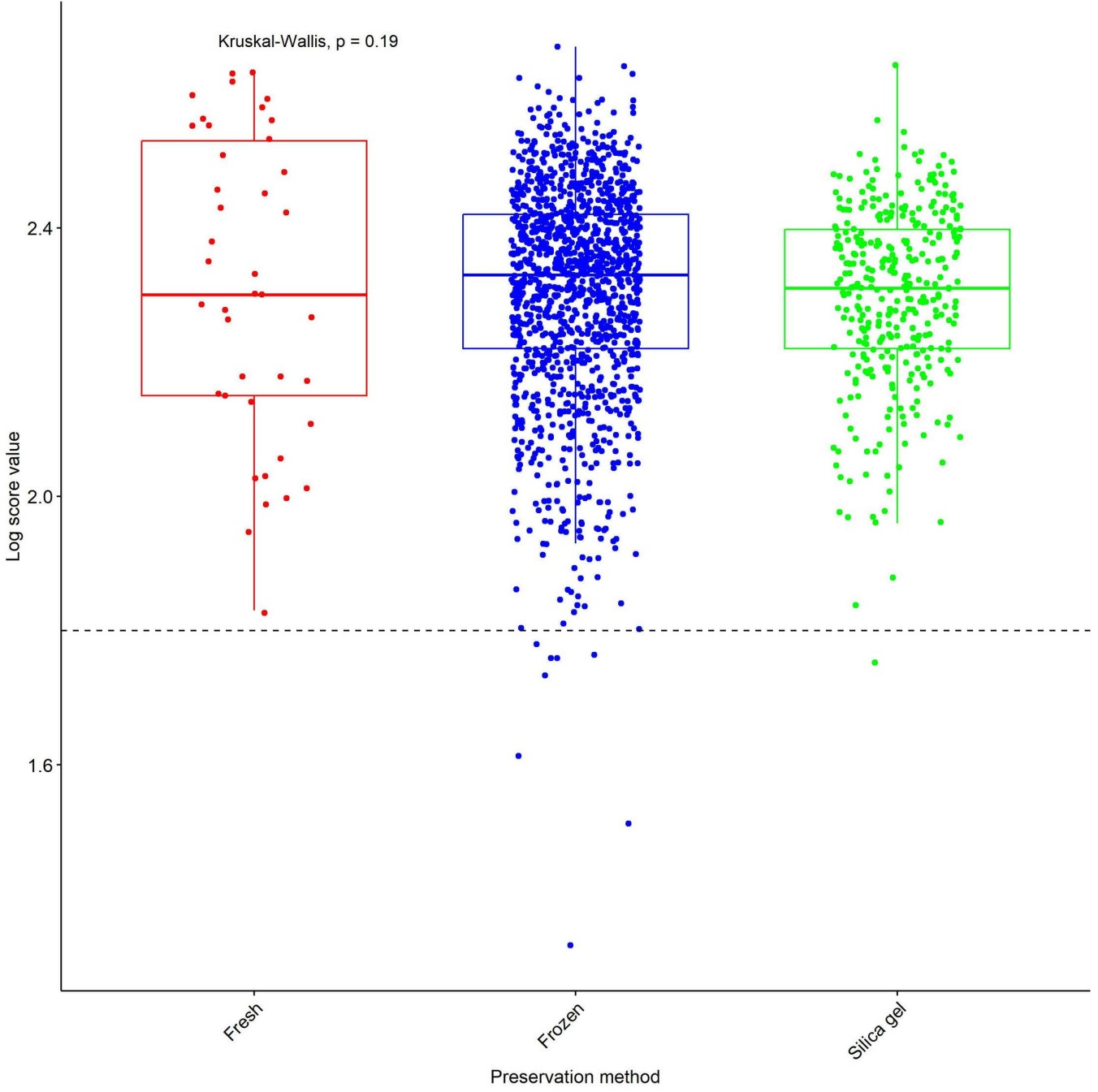

**Fig 3. Log Score Value (LSVs) obtained after MSP reference database query with MS spectra based on method of preservation.** Horizontal dashed lines represent the cut-off point for reliable identification (LSV > 1.8) [5].

the type of rigor required for effective entomological surveillance and the obvious benefit of using a highly sensitive tool such as MALDI-TOF MS. The spectra obtained from the head and thorax can simultaneously measure several entomological parameters if libraries are created for that purpose. In different studies, the head and thorax spectral pattern has been successfully used for species identification [5], pathogens infection status determination [21], endosymbionts [22] and age-grading [26].

More recently, a proof-of-concept study demonstrated that spectral patterns from the head and thorax were used for discriminating between resistant and susceptible mosquitoes [27]. Here, we propose using the head and thorax spectral for all future MALDI TOF MS spectra acquisition for malaria vector profiling. To make this technology accessible to all stakeholders, including national public health agencies and partners responsible for vector surveillance, it is essential to harmonize protocols and methodologies across research groups and institutions. Standardizing approaches will enable the development of a shared platform, ensuring consistency, comparability, and broader adoption of this technology in national malaria control programs.

In conclusion, the proposed work flow enables the creation of in-house MALDI-TOF MS libraries with relative ease (protocol and associated video are attached as supplementary files). MALDI TOF MS can be used to robustly identify mosquito species. The high accuracy and ability to distinguish complex species and visualise species relationships make this method valuable in entomological monitoring of primary Afrotropical malaria vectors and can be expanded to include other mosquito disease vectors.

## Supporting information

**S1 File. MALDI-TOF MS library development for insects.** This PDF contains the detailed step-by-step protocol for constructing MALDI-TOF MS reference database for insect identification.
(PDF)

## Acknowledgments

We are grateful to the Scientific and technical teams at the Centre for Geographic Medicine Research Coast, Kilifi for help in design and implementation of this work. Many thanks to the technical and field staff (Festus Yaa, Gabriel Nzai, Robert Mwakesi and Martha Muturi) who devoted their time and assisted in the field collection of samples. This paper has been published with the permission of the Director of the Kenya Medical Research Institute (KEMRI).

**Associated content**

- Mosquito Database Creation Using MALDI TOF MS video demonstration available on YouTube (https://youtu.be/btS0hsx9yNw). The video shows the experimental steps of MALDI TOF MS database creation.

## Author contributions

**Conceptualization:** Jonathan Karisa, Martin Rono, Joseph Mwangangi, Marta Maia.

**Data curation:** Jonathan Karisa, Mercy Tuwei, Marta Maia.

**Formal analysis:** Jonathan Karisa, Marta Maia.

**Funding acquisition:** Martin Rono, Philip Bejon, Joseph Mwangangi, Marta Maia.

**Investigation:** Jonathan Karisa, Mercy Tuwei, Kelly Ominde, Brian Bartilol, Zedekiah Ondieki, Haron Musani, Martin Rono, Charles Mbogo, Philip Bejon, Joseph Mwangangi, Caroline Wanjiku, Marta Maia.

**Methodology:** Jonathan Karisa, Mercy Tuwei, Kelly Ominde, Brian Bartilol, Zedekiah Ondieki, Haron Musani, Martin Rono, Charles Mbogo, Philip Bejon, Joseph Mwangangi, Caroline Wanjiku, Marta Maia.

**Project administration:** Martin Rono, Joseph Mwangangi, Caroline Wanjiku, Marta Maia.

**Resources:** Charles Mbogo, Joseph Mwangangi, Caroline Wanjiku, Marta Maia.

**Supervision:** Martin Rono, Charles Mbogo, Philip Bejon, Joseph Mwangangi, Caroline Wanjiku, Marta Maia.

**Writing – original draft:** Jonathan Karisa.

**Writing – review & editing:** Mercy Tuwei, Kelly Ominde, Brian Bartilol, Zedekiah Ondieki, Haron Musani, Martin Rono, Charles Mbogo, Philip Bejon, Joseph Mwangangi, Caroline Wanjiku, Marta Maia.

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
