## [Decision Letter · Decision Letter 0]

26 May 2025

Dear Dr. Karisa,

Thank you for submitting your manuscript to PLOS ONE. After careful consideration, we feel that it has merit but does not fully meet PLOS ONE’s publication criteria as it currently stands. Therefore, we invite you to submit a revised version of the manuscript that addresses the points raised during the review process.

The procedures and results reported in this manuscript have largely been been published previously (https://doi.org/10.12688/wellcomeopenres.18982.2) and the added value of this article appears somewhat limited.

In the abstract section, the authors wrote "Libraries are built in-house for institutional usage, although a multi-user platform with sharing of spectra and data would be attractive, it is not yet available." This sentence is erroneous and should revised, providing references to published studies that strived to develop a free identification service available online (https://doi.org/10.1371/journal.pone.0305167), a free open-source data analysis pipeline and share mass spectral data (https://doi.org/10.1186/s13071-024-06655-1).

I would also strongly recommend to the authors to deposit the mass spectra on a public repository to facilitate the widespread uptake of MALDI-TOF MS for mosquito species identification.

We look forward to receiving your revised manuscript.

Kind regards,

Dina Aboelsoued, Ph.D.

Academic Editor

PLOS ONE

Journal Requirements:

2. We note you have not yet provided a protocols.io PDF version of your protocol and/or a protocols.io DOI. When you submit your revision, please provide a PDF version of your protocol as generated by protocols.io (the file will have the protocols.io logo in the upper right corner of the first page) as a Supporting Information file. The filename should be S1_file.pdf, and you should enter “S1 File” into the Description field. Any additional protocols should be numbered S2, S3, and so on. Please also follow the instructions for Supporting Information captions [https://journals.plos.org/plosone/s/supporting-information#loc-captions]. The title in the caption should read: “Step-by-step protocol, also available on protocols.io.”

Please assign your protocol a protocols.io DOI, if you have not already done so, and include the following line in the Materials and Methods section of your manuscript: “The protocol described in this peer-reviewed article is published on protocols.io (https://dx.doi.org/10.17504/protocols.io.[...]) and is included for printing purposes as S1 File.” You should also supply the DOI in the Protocols.io DOI field of the submission form when you submit your revision.

If you have not yet uploaded your protocol to protocols.io, you are invited to use the platform’s protocol entry service [https://www.protocols.io/we-enter-protocols] for doing so, at no charge. Through this service, the team at protocols.io will enter your protocol for you and format it in a way that takes advantage of the platform’s features. When submitting your protocol to the protocol entry service please include the customer code PLOS2022 in the Note field and indicate that your protocol is associated with a PLOS ONE Lab Protocol Submission. You should also include the title and manuscript number of your PLOS ONE submission.

“MM - Investment 057212 - Bill and Melinda Gates Foundation - USA

MR - FLR_R1_190497 - The Royal Society FLAIR fellowship grant- UK”

4. We noted in your submission details that a portion of your manuscript may have been presented or published elsewhere. “Utility of MALDI-TOF MS for determination of species identity and blood meal sources of primary malaria vectors on the Kenyan coast” in Wellcome Open Research (10.12688/wellcomeopenres.18982.2).“

**Additional Editor Comments:**

The procedures and results reported in this manuscript have largely been been published previously (https://doi.org/10.12688/wellcomeopenres.18982.2) and the added value of this article appears somewhat limited.

In the abstract section, the authors wrote "Libraries are built in-house for institutional usage, although a multi-user platform with sharing of spectra and data would be attractive, it is not yet available." This sentence is erroneous and should revised, providing references to published studies that strived to develop a free identification service available online (https://doi.org/10.1371/journal.pone.0305167), a free open-source data analysis pipeline and share mass spectral data (https://doi.org/10.1186/s13071-024-06655-1).

I would also strongly recommend to the authors to deposit the mass spectra on a public repository to facilitate the widespread uptake of MALDI-TOF MS for mosquito species identification.

Reviewers' comments:

Reviewer's Responses to Questions

**Comments to the Author**



Reviewer #1: Yes

Reviewer #2: Yes

2. Has the protocol been described in sufficient detail?

To answer this question, please click the link to protocols.io in the Materials and Methods section of the manuscript (if a link has been provided) or consult the step-by-step protocol in the Supporting Information files.

Reviewer #1: Yes

Reviewer #2: Yes

3. Does the protocol describe a validated method?

Reviewer #1: Yes

Reviewer #2: Yes

4. If the manuscript contains new data, have the authors made this data fully available?

Reviewer #1: No

Reviewer #2: Yes

**5. Is the article presented in an intelligible fashion and written in standard English?**

Reviewer #1: Yes

Reviewer #2: Yes

Reviewer #1: In this protocol paper, the authors present a method to develop reference databases for Anopheles species identification with MALDI-TOF MS in-house. The procedures and results reported in this manuscript have largely been been published previously (https://doi.org/10.12688/wellcomeopenres.18982.2) and the added value of this article appears somewhat limited. In the abstract section, the authors wrote "Libraries are built in-house for institutional usage, although a multi-user platform with sharing of spectra and data would be attractive, it is not yet available." This sentence is erroneous and should revised, providing references to published studies that strived to develop a free identification service available online (https://doi.org/10.1371/journal.pone.0305167), a free open-source data analysis pipeline and share mass spectral data (https://doi.org/10.1186/s13071-024-06655-1). I would also strongly recommend to the authors to deposit the mass spectra on a public repository to facilitate the widespread uptake of MALDI-TOF MS for mosquito species identification.

Reviewer #2: The Lab Protocol titled " Making MALDI-TOF MS for entomological parameters accessible: A practical guide for in-house library creation was interesting and provided valuable information for readers. It will be publishable

**Do you want your identity to be public for this peer review?** For information about this choice, including consent withdrawal, please see our Privacy Policy

Reviewer #1: No

Reviewer #2: No

---

## [Author Response · Author response to Decision Letter 1]

7 Jul 2025

Comments and response

The manuscript and associated figures and supplementary documents have been edited and aligned as per the PLOS ONE requirements

2. We note you have not yet provided a protocols.io PDF version of your protocol and/or a protocols.io DOI. When you submit your revision, please provide a PDF version of your protocol as generated by protocols.io (the file will have the protocols.io logo in the upper right corner of the first page) as a Supporting Information file. The filename should be S1_file.pdf, and you should enter “S1 File” into the Description field. Any additional protocols should be numbered S2, S3, and so on. Please also follow the instructions for Supporting Information captions [https://journals.plos.org/plosone/s/supporting-information#loc-captions]. The title in the caption should read: “Step-by-step protocol, also available on protocols.io.” Please assign your protocol a protocols.io DOI, if you have not already done so, and include the following line in the Materials and Methods section of your manuscript: “The protocol described in this peer-reviewed article is published on protocols.io (https://dx.doi.org/10.17504/protocols.io.[...]) and is included for printing purposes as S1 File.” You should also supply the DOI in the Protocols.io DOI field of the submission form when you submit your revision.

If you have not yet uploaded your protocol to protocols.io, you are invited to use the platform’s protocol entry service [https://www.protocols.io/we-enter-protocols] for doing so, at no charge. Through this service, the team at protocols.io will enter your protocol for you and format it in a way that takes advantage of the platform’s features. When submitting your protocol to the protocol entry service please include the customer code PLOS2022 in the Note field and indicate that your protocol is associated with a PLOS ONE Lab Protocol Submission. You should also include the title and manuscript number of your PLOS ONE submission.

The step-by-step protocol of the process here briefly described, is also available on protocols.io (dx.doi.org/10.17504/protocols.io.n2bvjdy3xvk5/v1) and is included as a supplementary file with this article

3. Thank you for stating the following financial disclosure: “MM - Investment 057212 - Bill and Melinda Gates Foundation – USA. MR - FLR_R1_190497 - The Royal Society FLAIR fellowship grant- UK”Please state what role the funders took in the study. If the funders had no role, please state: "The funders had no role in study design, data collection and analysis, decision to publish, or preparation of the manuscript." If this statement is not correct you must amend it as needed. Please include this amended Role of Funder statement in your cover letter; we will change the online submission form on your behalf.

The funders had no role in study design, data collection and analysis, decision to publish, or preparation of the manuscript

4. We noted in your submission details that a portion of your manuscript may have been presented or published elsewhere. “Utility of MALDI-TOF MS for determination of species identity and blood meal sources of primary malaria vectors on the Kenyan coast” in Wellcome Open Research (10.12688/wellcomeopenres.18982.2). “Please clarify whether this [conference proceeding or publication] was peer-reviewed and formally published. If this work was previously peer-reviewed and published, in the cover letter please provide the reason that this work does not constitute dual publication and should be included in the current manuscript.

Utility of MALDI-TOF MS for determination of species identity and blood meal sources of primary malaria vectors on the Kenyan coast” in Wellcome Open Research (10.12688/wellcomeopenres.18982.2) is a peer reviewed publication. However, the current submission is a methodology manuscript detailing how to build a MALDI-TOF MS library for entomological samples and explaining the various technical steps analytical steps supported by a protocol. Although we present a case study on mosquitoes, this manuscript is relevant to this working on different insect classes. We detail this approach using data from a previously published article on Afro tropical malaria vectors. The present submission differs in scope and objectives from what has been previously published earlier which demonstrated the method’s ability to identify vectors compared to gold standard. The earlier paper has been referenced. The data from the previous published work is shown in this manuscript only as a case study to elucidate how to build the libraries and employ this technology.

All the data related to this publication has been deposited in the Harvard Dataverse. This includes the dataset, and MALDI-TOF MS Main Spectral Profiles (Libraries) for species identification (https://doi.org/10.7910/DVN/VYQFNO37)

All the references are per as the PLOS one guidelines

Additional Editor Comments:

The procedures and results reported in this manuscript have largely been been published previously (https://doi.org/10.12688/wellcomeopenres.18982.2) and the added value of this article appears somewhat limited.

The current submission is a methodology manuscript detailing how to build a MALDI-TOF MS library for entomological samples and explaining the various technical steps analytical steps supported by a protocol. Although we present a case study on mosquitoes, this manuscript is relevant to this working on different insect classes. We detail this approach using data from a previously published article on Afro tropical malaria vectors. The present submission differs in scope and objectives from what has been previously published earlier which demonstrated the method’s ability to identify vectors compared to gold standard. The earlier paper has been referenced. The data from the previous published work is shown in this manuscript only as a case study to elucidate how to build the libraries and employ this technology.

In the abstract section, the authors wrote "Libraries are built in-house for institutional usage, although a multi-user platform with sharing of spectra and data would be attractive, it is not yet available." This sentence is erroneous and should revised, providing references to published studies that strived to develop a free identification service available online (https://doi.org/10.1371/journal.pone.0305167), a free open-source data analysis pipeline and share mass spectral data (https://doi.org/10.1186/s13071-024-06655-1).

The sentence has been revised to read “Libraries are built in-house for institutional usage, although a multi-user platform with sharing of spectra and data would be attractive. Only a few studies have strived to make their libraries publicly available”

Moreover, the MALDI-TOF MS Main Spectral Profiles (Libraries) for species identification related to this publication has been deposited in public repository - Harvard Dataverse (https://doi.org/10.7910/DVN/VYQFNO37)

I would also strongly recommend to the authors to deposit the mass spectra on a public repository to facilitate the widespread uptake of MALDI-TOF MS for mosquito species identification.

The MALDI-TOF MS Main Spectral Profiles (Libraries) for species identification related to this publication has been deposited in public repository i.e. the Harvard Dataverse (https://doi.org/10.7910/DVN/VYQFNO37) for any one with MALDI-TOF MS equipment to adapt and update for their use

---

## [Editor Report · Decision Letter 1]

17 Jul 2025

Dear Dr. Karisa

Thank you for submitting your manuscript to PLOS ONE. After careful consideration, we feel that it has merit but does not fully meet PLOS ONE’s publication criteria as it currently stands. Therefore, we invite you to submit a revised version of the manuscript that addresses the points raised during the review process.

We look forward to receiving your revised manuscript.

Kind regards,

Dina Aboelsoued, Ph.D.

Academic Editor

PLOS ONE

Journal Requirements:

Additional Editor Comments:

The present submission demonstrated the method’s ability to identify vectors compared to gold standard. As the earlier paper has been referenced and used to show how to build the libraries and employ this technology.

Authors should shorten the details shown in the current manuscript referring to other published work

---

## [Author Response · Author response to Decision Letter 2]

20 Jul 2025

Comments and response

1. If the reviewer comments include a recommendation to cite specific previously published works, please review and evaluate these publications to determine whether they are relevant and should be cited.

Response – In the previous review, the reviewers recommended two articles:

- (https://doi.org/10.1371/journal.pone.0305167), a free open-source data analysis pipeline and share mass spectral data

- (https://doi.org/10.1186/s13071-024-06655-1).

Upon reviewing these publications, we noted that the methods used differ from ours. Specifically, both studies employed R and Python scripts for post-acquisition spectral processing. In contrast, our study utilized the MALDI-Biotyper® software for spectral processing and analysis. None of these reference have been included in our manuscript. Regarding the recommendation to deposit mass spectra in a public repository, we confirm that all relevant data associated with this study have been deposited in the Harvard Dataverse. This includes both the raw dataset and the MALDI-TOF MS Main Spectral Profiles (MSPs) used for species identification. The data are accessible via the following DOI: https://doi.org/10.7910/DVN/VYQFNO37

Response – Following your suggestion on the references cited in our manuscript, we conducted a thorough review of all the references included in the manuscript. We confirm that none of the cited references appear to have been retracted, and all remain accessible through their respective journals or digital repositories. To the best of our knowledge, all references are from peer-reviewed and meet PLOSOne journal requirements. If the editors or reviewers have any specific concerns regarding individual references, we would be happy to address them promptly.

3. The present submission demonstrated the method’s ability to identify vectors compared to gold standard. As the earlier paper has been referenced and used to show how to build the libraries and employ this technology. Authors should shorten the details shown in the current manuscript referring to other published work

Response – The manuscript, especially the methods section, has been revised and the relevant published work and step-by-step protocol deposited in protocol.io has been referenced.

---

## [Editor Report · Decision Letter 2]

5 Aug 2025

Making MALDI-TOF MS for entomological parameters accessible: A practical guide for in-house library creation

PONE-D-25-21203R2

Dear Mr. Jonathan K Karisa

We’re pleased to inform you that your manuscript has been judged scientifically suitable for publication and will be formally accepted for publication once it meets all outstanding technical requirements.

Kind regards,

Dina Aboelsoued, Ph.D.

Academic Editor

PLOS ONE

---

## [Editor Report · Acceptance letter]

PONE-D-25-21203R2

PLOS ONE

Dear Dr. Karisa,

I'm pleased to inform you that your manuscript has been deemed suitable for publication in PLOS ONE. Congratulations! Your manuscript is now being handed over to our production team.

Kind regards,

on behalf of

Dr. Dina Aboelsoued

Academic Editor

PLOS ONE